# Engineering surface state density of monolayer CVD grown 2D MoS$_2$ for enhanced photodetector performance

Gowtham Polumati[1], Chandra Sekhar Reddy Kolli[1], Andres de Luna Bugallo[2]*, Parikshit Sahatiya[1,3]*

1 Department of Electrical and Electronics Engineering, BITS Pilani, Hyderabad Campus, Hyderabad, India, 2 Materials Center for Sustainable Energy & Environment, Birla Institute of Technology and Science Pilani, Hyderabad Campus, Hyderabad, India, 3 Centro de Física Aplicada y Tecnología Avanzada, Universidad Nacional Autónoma de México, A.P. 1–1010, Querétaro, Qro., México

* aluna@fata.unam.mx (ALB); parikshit@hyderabad.bits-pilani.ac.in (PS)

**Data Availability Statement:** All relevant data are within the manuscript and its Supporting Information files.

## Abstract

This study demonstrates the effect of nitrogen doping on the surface state densities (Nss) of monolayer MoS$_2$ and its effect on the responsivity and the response time of the photodetector. Our experimental results shows that by doping monolayer MoS$_2$ by nitrogen, the surface state (Nss) increases thereby increasing responsivity. The mathematical model included in the paper supports the relation of photocurrent gain and its dependency on trap level which states that the increasing the trap density increases the photocurrent gain and the same is observed experimentally. The experimental results at room temperature revealed that nitrogen doped MoS$_2$ have a high N$_{SS}$ of 1.63 X 10$^{13}$ states/m$^2$/eV compared to undoped MoS$_2$ of 4.2 x 10$^{12}$ states/m$^2$/eV. The increase in Nss in turn is the cause for rise in trap states which eventually increases the value of photo responsivity from 65.12 A/W (undoped MoS$_2$) to 606.3 A/W (nitrogen doped MoS$_2$). The response time calculated for undoped MoS$_2$ was 0.85 sec and for doped MoS$_2$ was 0.35 sec. Finally, to verify the dependence of surface states on the responsivity, the surface states were varied by varying temperature and it was observed that upon increment in temperature, the surface states decreases which causes the responsivity values also to decrease.

## Introduction

Photodetectors are the devices that converts incident light signal into electrical signal. There are many factors which contribute to the improved performance of any photodetector such as defects, traps and surface states [1]. It is well known that the presence of traps has a tendency to increase the carrier lifetime, leading to reduced recombination and enhanced conductivity in devices [2, 3]. This phenomenon plays a considerable role in enhancing the responsivity of the device. There are various ways to introduce traps, like creating defects during synthesis process, carrier generation and recombination, radiation, tunnelling, varying density of states (Nss) upon doping [4–6]. Although there are many ways to vary these surface states, out of

**Funding:** The author(s) received no specific funding for this work.

**Competing interests:** NO authors have competing interests

which few are incomplete covalent bonds, discontinuities between semiconductor and metal, chemical reactions during fabrication process and finally the doping. Out of many, engineering the density of surface states through controlled doping, is preferred as it involves with considerable variation of atomic layers that are on the surface of the device [7]. The surface states are termed as electronic states that lies on the surface of any solid material. These surface states are found only on atomic layers closest to the surface of solid material due to sharp transition of solid material to its surface [8]. Which means for any solid material that reaches to termination of its surface, the material electronic band structure will change from bulk to the vacuum creating electronic states that are formed at transition and these electronic states are generally found at atomic layers that are close to the surface [8–11]. For a photodetectors, upon light matter interaction these surface states play a crucial role in converting light into corresponding current. Also, by doping the device with external dopants, these surface states which were on the atomic layers of solid device are replaced by the dopant atoms consequently enhancing the device performance [12, 13]. Conventional photodetectors suffer from a lot of drawbacks like low absorption, defects that are formed during fabrication process, poor responsivity, expensive synthesis procedures which makes them unsuitable in many applications in demand [14–17]. Hence there is a need to search for new material which will have suitable properties that can serve the purpose, and one such materials are 2D materials. In our previous work, the study of nitrogen-doping of MoS$_2$ and the effect of substrate was performed [18]. However, the investigation into the underlying reasons for the improved responsivity resulting from doping was not thoroughly explored. The explanation was primarily limited to the electric field generation at the interface. Nevertheless, it is crucial to consider the influence of surface states in order to gain a comprehensive understanding of the transport mechanisms and the enhanced responsivity associated with doping, this aspect has been scarcely investigated in the existing literature.

The current work demonstrates the systematic procedure in enhancing responsivity of monolayer MoS$_2$ photodetector device with introduction of traps by varying its surface states upon effective nitrogen doping. By doping, the surface states are varied experimentally from 4.2 x 10$^{12}$ states/m$^2$/eV (undoped MoS$_2$/n-Si device) to 1.63 X 10$^{13}$ states/m$^2$/eV (nitrogen doped MoS$_2$/n-Si device). Upon variation in surface states, the traps are created and act as defect centres or impurities that captures excess carriers just below the conduction band [19]. As a result, there exists quite a good number of carriers below conduction band. With external bias applied, the carriers captured by traps provide additional current to that of the actual current. Due to this for nitrogen doped MoS$_2$/n-Si device, the photogenerated current is higher than undoped MoS$_2$/n-Si device. Therefore, the responsivity of nitrogen doped MoS$_2$/n-Si device (606.3 A/W) is almost 10 times compared to undoped MoS$_2$/n-Si device (65.12 A/W). Furthermore, the response time is observed to be less for high responsivity device (doped MoS$_2$). To investigate the influence of surface states on the photodetector responsivity, measurements were conducted at various temperatures, considering the strong temperature-dependent nature of surface states. It was observed that with gradual increase in device temperature the density of surface states (Nss) decreases. This is attributed to that fact that with increase in device temperature, carriers present in trap level below conduction band will acquire sufficient energy and are excited to conduction band. Consequently, trap level is empty of excess carriers resulting a decrease in surface states Nss which in turn reduces the responsivity values. The present study of nitrogen doped MoS$_2$/n-Si photodetector device and enhancing its responsivity with varying surface states Nss at different temperatures gives many interesting and conceptual challenges for many researchers in the field of 2D materials for different device applications.

## Materials and methods

### Synthesis of Nitrogen Doped MoS$_2$ using CVD

Molybdenum trioxide (MoO$_3$) and sulfur (S) powder of 140 mg and 200 mg respectively are kept in two separate aluminium oxide (Al$_2$O$_3$) boats. Si wafer is treated with RCA cleaning and then the same n-Si wafer having dimensions of (1 cm × 2 cm) was kept facing down on to the Al$_2$O$_3$ boat containing MoO$_3$.Once the monolayer MoS2 flakes are grown, their morphology was confirmed by FESEM and then the same sample is introduced into CVD for nitrogen doping. The sample consisting of MoS2 flakes is kept in one of the zones in CVD and then nitrogen is being purged at 40 sccm. The complete synthesis procedure and the doping process along with schematic can be found in our previous paper reported from our lab [18].

## Results and discussion

CVD method was used to grow large area nitrogen doped MoS$_2$ crystals. The complete synthesis and device characterization of FESEM, HRTEM, XPS, Raman, PL and UPS are shown in **S1–S3** **Figs** of Supplementary Information (SI) respectively. For the electrical characterization current–voltage (I-V) measurements were performed at room temperature. I-V characteristics for both undoped MoS$_2$/n-Si device and nitrogen doped MoS$_2$/n-Si device having the device dimensions (1 cm × 2 cm) were examined as shown in **Fig 1A) and 1B)**.

To further analyze the charge transport phenomenon of this MoS$_2$/Si junction, thermionic emission analysis was applied [20] and can be expressed as

$$I = I_O exp\left(\frac{q(V - IR)}{\eta KT}\right) \tag{1}$$

$$I_o = AA^* T^2 \exp\left(\frac{-q\Phi_B}{KT}\right), A^* = \frac{4\pi q m^* k^2}{h^2} \tag{2}$$

$$\eta = \frac{q}{KT}\frac{dV}{d(\ln I)} \tag{3}$$

Where, I$_0$, K, q, h are reverse saturation current, Boltzmann constant, electron charge and Planck's constant respectively. η is called an ideality factor indicates the deviation of the device performance from ideal diode. A is junction area, T is ambient temperature, R is series resistance, A* is Richardson constant which is theoretically estimated to be $0.70 \times 10^{-6}$ A cm$^{-2}$ K$^{-2}$ for MoS$_2$, considering effective mass m* = 0.27 m$_0$ [21]. Φ$_B$ is the barrier height at zero bias,

The ideality factor η and Schottky barrier height Φ were calculated by linear fit of natural log plot of voltage versus current. The ideality factor is generally taken to be unity in ideal condition and it generally deviates from ideal value 1 which can be anticipated due to the defects introduced in MoS$_2$ during synthesis process, like chemical reaction, incomplete covalent bonds and doping, and existence of surface states which results in many unwanted current at the interface [22]. In order to analyse the MoS$_2$/Si interface quality of both doped and undoped devices, voltage dependence of η was measured and utilized to examine the density of interface states.

Under equilibrium, Density of surface states can be analysed by the equation shown

$$N_{ss} = \frac{\epsilon_i}{t_i q^2}(\eta - 1) - \frac{\epsilon_{sc}}{wq^2} \tag{4}$$

where ε$_i$ is relative dielectric constant and t$_i$ is the thickness of the interfacial layer. The value

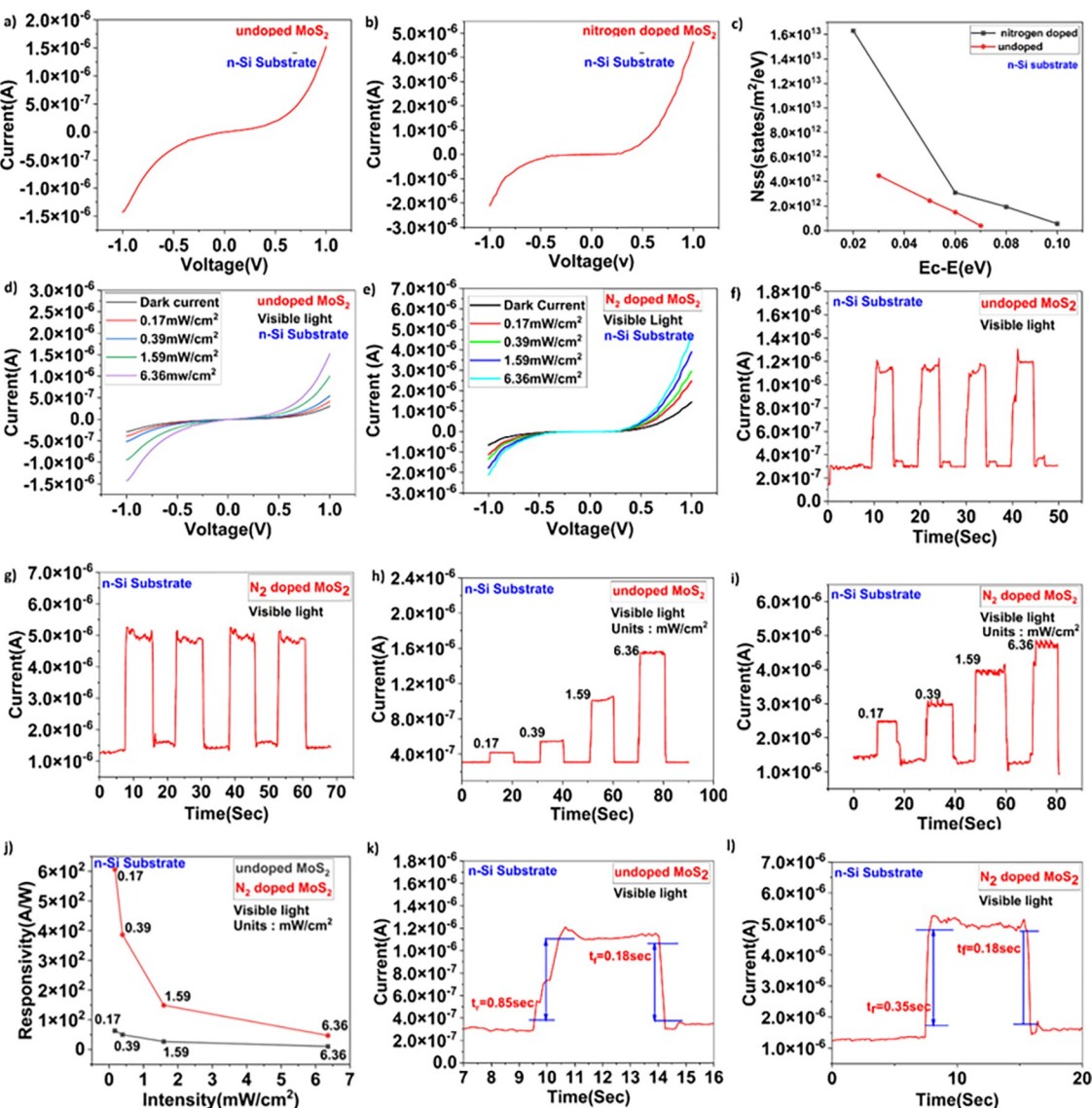

**Fig 1.** a) Graph showing I-V for undoped MoS$_2$/n-Si device. b) Graph showing I-V for nitrogen doped MoS$_2$/n-Si device. c) Nss of both nitrogen doped MoS$_2$/n-Si junction and undoped MoS$_2$/n-Si junction. d-e) Graph showing I-V's for both undoped and doped MoS$_2$/n-Si device. f-g) Graph showing temporal response for constant light intensity for both nitrogen doped and undoped MoS$_2$/n-Si device. h-i) Graph showing temporal response for variable light intensity for both nitrogen doped and undoped MoS$_2$/n-Si device. j) responsivity of nitrogen doped and undoped MoS$_2$/n-Si junction. k-l) rise time for both undoped and doped MoS$_2$/n-Si device.

of $\varepsilon_i$ was assumed to be permittivity of free space and thickness to be 0.65nm. $\varepsilon_{sc}$ and W are the relative dielectric constant and depletion width respectively. The value of $\varepsilon_{sc}$ is 3.7 [23] for MoS$_2$ and width of the depletion region to be 60 x 10$^{-6}$ m. Using Eq (4) and the bias voltage dependence of Nss, Nss (V) can be obtained by following equation

$$E_C - E = \Phi_B - qV \tag{5}$$

**Fig 1(C)** shows the graph of Nss (v) upon $E_C - E$ for MoS$_2$/n-Si(both nitrogen doped and undoped) indicates that the nitrogen doped MoS$_2$/n-Si has higher density of states 5.3 X 10$^{12}$ states/m$^2$/eV when compared to undoped MoS$_2$/n-Si 4.2 X 10$^{12}$ states/m$^2$/eV. As the surface

states are atomic layers that lies on the surface of any device, once the device is subjected to doping, the atomic layers on surface of the device are being occupied by dopant atoms which may affect device performance [19, 24, 25]. The Nss being high for nitrogen doped MoS$_2$/n-Si is due to the fact that in the fabrication process involving nitrogen doping, the vacancies or defect centres that are on the surface of the MoS$_2$/n-Si device are being occupied by dopant atoms which leads to variation in density of surface states. The variation in density of surface states further leads to trap creations and these traps play a major role in capturing excess carriers well below the conduction band and these carriers are high in number for nitrogen doped MoS$_2$/n-Si device when compared to undoped MoS$_2$/n-Si device. Further, it also leads to the increment in the carrier lifetime. As a result upon light illumination, for nitrogen doped MoS$_2$/n-Si the carrier lifetime and the carrier generation is more than the undoped MoS$_2$/n-Si junction.

When n-MoS$_2$ (doped and undoped) and n-Si junction device is subjected to external light illumination, and if the energy of the incident light is higher than the energy bandgap of the n-MoS$_2$, carrier generation happens as a result electron -hole pairs are created. The as generated carriers are segregated by the built-in electric field and are received at respective metal electrodes, which results in improved photocurrent. The magnitude of the photocurrent generated upon external light illumination depends on the strength of the built-in electric field and Schottky barrier height.

Photodetection experiment was carried out for both undoped and nitrogen doped MoS$_2$/n-Si device to calculate various parameters like Responsivity, Detectivity and EQE, Graph **d) and e)** of **Fig 1** indicates I-V's for both undoped and nitrogen doped MoS$_2$/n-Si device wherein it was observed that device currents are increasing with increase in incident light intensity. Similarly graphs **f)** and **g) of Fig 1** shows the temporal response (for constant incident light intensity) of both undoped and nitrogen doped MoS$_2$/n-Si device which represents that for constant intensity (0.39 mW/cm$^2$) of incident light the device is showing significant current when the light source is ON and reaching its initial value when the light source is off. Also, **Fig 1H)** and **1I)** shows the temporal response (for variable incident light intensity)of both undoped and nitrogen doped MoS$_2$/n-Si device for variable incident light intensities(0.17mW/cm$^2$, 0.39mW/cm$^2$, 1.59mW/cm$^2$, 6.36mW/cm$^2$) indicating that current for both nitrogen doped MoS$_2$/n-Si and undoped MoS$_2$/n-Si devices was increased with increase in intensity of incident light.

Responsivity, which is a major parameter and qualitative measure in analyzing any photodetector is defined as the photocurrent generated per unit area upon light incidence per unit power [26]. The Responsivity is given mathematically as follows

$$R_\lambda = \frac{I_\lambda}{A * P_\lambda} \qquad (6)$$

where I$_\lambda$, P$_\lambda$, A, $I_{dark}$, E and $\lambda$ are photogenerated current of the device, illumination power of source, active area of the device, dark current of the device, and charge of an electron, wavelength of incident source, respectively.

Furthermore, Responsivity of both nitrogen doped MoS$_2$ /n-Si and undoped MoS$_2$/n-Si devices are calculated as in **Fig 1J)** and it was examined that for nitrogen doped MoS$_2$/n-Si junction responsivity was 606.3 A/W and for undoped MoS$_2$/n-Si it was 65.12 A/W. It was noted that the responsivity of nitrogen doped MoS$_2$/n-Si device is almost 10 times higher than undoped MoS$_2$/n-Si junction. Rise time for both nitrogen doped MoS$_2$/n-Si junction and undoped MoS$_2$/n-Si are calculated using temporal response graphs of the photodetection experiment results. As the experimental results shows that for a nitrogen doped MoS$_2$/n-Si

device surface states and responsivity increases and rise time decreases compared to undoped device. According to existing literature, a high speed optoelectronic device should have high responsivity and low rise time [27]. Hence nitrogen doping is one best way to make a device suitable for high-speed optoelectronic applications. The calculated rise time for both nitrogen doped MoS$_2$/n-Si junction and undoped MoS$_2$/n-Si junction are 0.35 sec and 0.85 sec respectively and the same is shown in **Fig 1K and 1L).**

The energy band diagram of undoped MoS$_2$ (nitrogen doped MoS$_2$) and n-Si when isolated and contacted is shown in **Fig 2A)–2D)**. The detailed mechanism of charge transfer and carrier migration with corresponding band structure was explained in our previous reported paper from our lab [18]. When the nitrogen doped MoS$_2$ and p-Si forms a junction, the fermi level aligns. MoS$_2$ is intrinsically n type and when nitrogen is doped there is a suppression of the n type behavior of MoS$_2$. When coupled with n-Si, the electric field generation is highest among the combination of different substrates (both n and p-Si). The detailed explanation of such comparison can be found in a recent report from our lab [14]. Based on the study performed, the nitrogen doped MoS$_2$ coupled with n-Si gives the highest responsivity and the reason being the high electric field due to the large Schottky barrier height. But there is another factor that plays a crucial role in improving the responsivity other than the electric field and that is surface states. As can be seen in Fig 1C, the surface states vary upon the doping and hence it not only the electric field but also the surface states that play an important role in determining the overall responsivity.

The variation in the density of surface states occurs due to the fact that for nitrogen doped MoS$_2$/n-Si junction, all the vacancies or defect that lies on the surface of the MoS$_2$/n-Si device are being occupied by dopant (nitrogen) atoms. This doping creates a considerable variation in density of surface states [28–30]. As the surface states lie on the surface of the device, upon doping, these surface states are varied as a result the responsivity of the device is also varied accordingly [27].

By doping the monolayer MoS$_2$ with nitrogen, the surface states are introduced as doping predominantly is a surface phenomenon. The introduction of surface states introduces extra energy levels near the conduction band which increases the carrier lifetime thereby increasing the photocurrent gain and the responsivity.

To fully understand the dependence of density of surface states on the performance of the fabricated device, the surface states were modulated by varying temperature. This is due to the fact that the density of surface states is a very strong function of temperature (Eq 4). For nitrogen doped MoS$_2$/n-Si device, experiment was carried out to examine the variation in density of surface states upon temperature. It is clear from the above Eqs 1),2),3) and 4) that as temperature is increased, it was observed a proportionate increase in device current with increase in device temperature and the I-Vs of the device were shown in **Fig 3A)**. Barrier height and reverse saturation current were also measured by varying (increasing) the device temperature and it was noticed that barrier height $\Phi_B$ decreases and reverse saturation current I$_o$ increases and the same is shown in **Fig 3B) and 3C)**. Furthermore, to assess the impact of temperature on the nitrogen-doped MoS$_2$/n-Si device, experiments were conducted to calculate the density of states (Nss). It was observed that the Nss curve exhibited a decreasing trend with increasing temperature, as depicted in **Fig 3D)**. The decrease in Nss trend with increase in temperature can be attributed to decrease in trap states [27]. As the device temperature increases, external heat is applied, leading to the elimination of trap energy levels that capture excess carriers. This occurs because the carriers, upon receiving external heat energy, gain sufficient energy to excite deep into the conduction band instead of being trapped at lower energy levels. As a result, all the carriers that are in trap level are being excited to conduction band resulting in

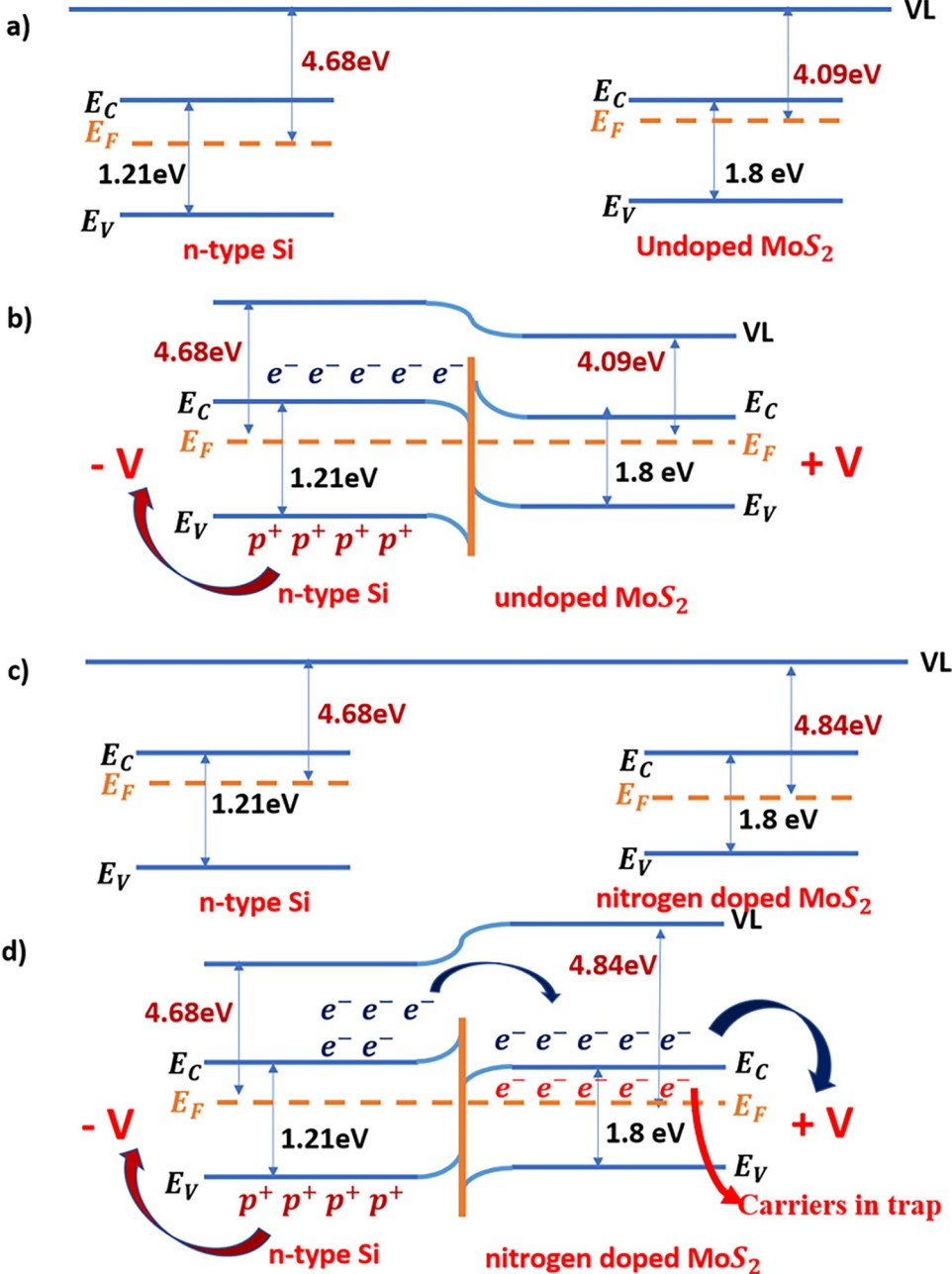

**Fig 2.** a-b) Schematic representing band diagram of n-type Si substrate and n-type undoped MoS₂ both when isolated and when contacted. c-d) Band diagram of n-type Si substrate and n-type nitrogen doped MoS₂ both when isolated and contacted.

decrease in carriers at trap level which in turn is the major cause for decrease in Nss trend and responsivity as shown in **Fig 3E**).

To further validate the effect of surface states, a mathematical model was derived which gave the relation of conductivity change as a function of traps, which finally concludes that the presence of traps will enhance conductivity resulting in increased responsivity. The corresponding mathematical analysis and its detailed explanation is as follows

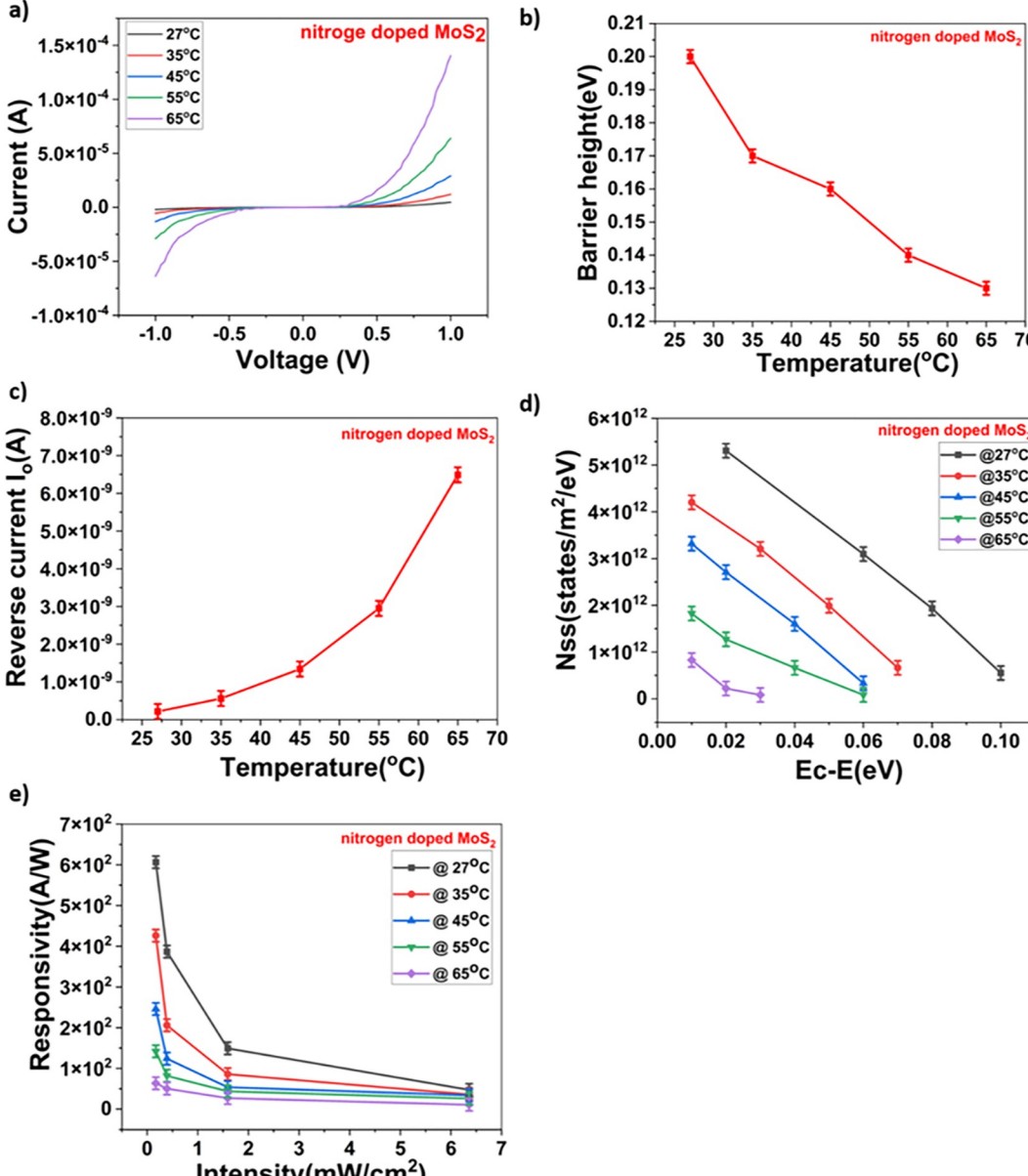

**Fig 3.** a) Graph showing different currents at different temperatures b) Graph showing decrease in barrier height with increase in temperature. c) Graph showing increase in reverse saturation current with increase in temperature. d)graph showing NSS at different temperatures. e) Responsivity at different temperatures.

### Mathematical model

The photodetector absorbs photons from incident light and generates corresponding electron hole pairs thereby causing considerable photocurrent. The photocurrent by definition is given by $I_{ph}$. Therefore, the continuity equation of a semiconductor to explain current conduction due to electrons [19] can be given by

$$\frac{\partial n}{\partial t} = \frac{\nabla J_n}{e} + G_n(x\,y\,z\,\lambda) - \nabla n / t_r \tag{7}$$

where n (cm$^{-3}$) represents concentration of electrons, J$_n$ (A cm$^{-2}$) current density of electrons, e is charge of electron, G$_n$ (cm$^{-3}$ s$^{-1}$) is the rate of carrier generated upon illumination, $\Delta$n (cm$^{-3}$) is concentration of electrons at nonequilibrium, and t$_r$ (s) is carrier recombination time. Finally, at steady state, $\partial$n/$\partial$t = 0 and $\nabla$J$_n$/e = 0, so G$_n$ = $\Delta$n/t$_r$. which implies

$$\nabla h = \nabla n = G_n(x\,y\,z\,\lambda)t_r \tag{8}$$

where $\Delta$n (cm$^{-3}$) is electron concentration at nonequilibrium. G$_n$(x, y, z, $\lambda$) = $\varepsilon$(x, y, z)n$_0$$\lambda$/hC, where h is Planck's constant, C denotes velocity of light in vacuum, and $\varepsilon$ (cm$^{-3}$) is quantum efficiency. It should be noted that when light is being incident on the device, for each photon that is absorbed by the device there are carriers generated and an electron hole pair is created. Considering $\gamma$ as the total quantum efficiency of the semiconductor device.

The average generation rate will be given by

$$G_n(x, y\ z, , \lambda) = (\gamma n_0 \lambda)/h\ C\ WLD.$$

$$G_n(x\,y\,z\,\lambda) = \varepsilon(x\,y\,z)\frac{n_o\lambda}{hc}$$

$$G_n(x\,y\,z\,\lambda) = \frac{\gamma n_o\lambda}{hc(WLD)} \tag{9}$$

The increased conductivity due to illumination $\Delta\sigma$ (S cm$^{-1}$) is given by

$$\nabla\sigma = e(\mu_n\nabla n + \mu_h\nabla h) \tag{10}$$

Where, $\mu_n$ ($\mu_h$) (cm$^2$ V$^{-1}$ s$^{-1}$) represents mobility of electron (hole). Then, I$_{ph}$ can be given by

$$I_{Photo\ current} = \nabla\sigma EWD(e(\mu_n+\mu_h)\nabla h)\frac{V_d}{L}\left(\frac{\gamma n_o\lambda}{hc}\right)\left(\frac{t_r}{L\nabla h}\right) \tag{11}$$

where E = $\frac{V_d}{L}$ gives electric field in the photodetector. The pure photocurrent which measured in the absence of external light incidence is obtained by considering all the absorbed photons.

$$I_{Pure} = \frac{e\gamma n_o}{\lambda\text{hC}} \tag{12}$$

Then, the photocurrent gain can be obtained by,

$$\alpha = \frac{I_{Photo\ current}}{I_{Pure}} \tag{13}$$

$$\alpha = (\mu_n+\mu_h)\frac{V_d}{L^2}t_r$$

$$\alpha = \frac{I_{Photo\ current}}{I_{Pure}} = \frac{t_r}{t_t}$$

Therefore, the photocurrent gain $\alpha$ is the ratio of carrier recombination time t$_r$ and the transit time. t$_t$ = L$^2$/($\mu_h$ + $\mu_n$)V$_d$. From the above photocurrent gain, if t$_r$ >> t$_t$, for each electron that reaches positive terminal of the supplied electric field, an electron is released by negative terminal of electric field. when an electron is received at the positive electrode in an electric field in order to have charge neutrality. Thus maintaining continuous current until

recombination of carriers happens., thus producing a gain current I $_{photocurrent}$ that is too higher than I$_{pure}$. It is also noted that $\alpha$ is inversely proportional to L$^2$ and directly proportional to V$_d$, t$_r$, and $\mu_n$ ($\mu_p$). It's a proven fact from current research that mobility of 2D materials are much lower than those of conventional Si based semiconductors devices. which may be the ultimate cause of having higher recombination life times in the trap effect making them high gain devices [24].

The trap is often referred to defect centres that contains either of the carriers may it be electro or hole. For semiconductor of n-type, the traps are due to electrons of which the energy level is assumed to be close to conduction band [28]. As the light is being incident, the traps occur and the number of electrons in trap level increases. the amount of the electrons collected by the traps ($\Delta n_t$) (cm$^{-3}$) should be higher than $\Delta n$ in steady state. Considering the traps to be evenly spread with trap concentration of N$_t$ (cm$^{-3}$). According to theories of indirect recombination, the capture rate and emission rate of the electrons is r$_p$(N$_t$ − $\Delta n_t$)$\Delta n$, and r$_p$n$_1$$\Delta n_t$ respectively. where r$_p$ (cm$^3$ s$^{-1}$) represents electron capture coefficient and n$_1$ (cm$^{-3}$) is trap level. Then, the rates at which $\Delta n$ and $\Delta n_t$ changes over the time is written as,

$$\frac{d\Delta n}{dt} = G_n(x\,y\,z\,\lambda) - \frac{\nabla n}{t_r} + r_n n_1 \Delta n_t - r_n(N_t - \Delta n_t)\Delta n$$

$$\frac{d\Delta n}{dt} = r_n(N_t - \Delta n_t)\Delta n - r_n n_1 \Delta n_t \tag{14}$$

The four terms of the above equation are generation rate, recombination rate, emission rate, and capture rate. At steady state, d$\Delta n$/dt = 0 and d$\Delta n_t$/dt = 0, so $\Delta n$ = G$_n$t$_r$. Upon solving

$$\frac{\Delta n_t}{\Delta n} = \frac{N_t}{n_1 + G_n t_r}$$

Considering no trap states,

$$\Delta \sigma = \Delta n e \mu_n + \Delta p\, e \mu_p = q \Delta p (\mu_p + \mu_n).$$

Considering the trap effect, because of the charge neutrality, $\Delta p = \Delta n + \Delta n_t$, so we obtain

$$\nabla \sigma = e(\mu_n \nabla n + \mu_h \nabla h)$$

$$\nabla \sigma = e \nabla n (\mu_n + \mu_h)$$

$$\nabla \sigma = n e \mu_n + (\Delta n + \Delta n_t) e \mu_h$$

$$\nabla \sigma = \Delta n e \left( \mu_n + \left( 1 + \frac{N_t}{\Delta n + n_1} \right) \mu_p \right) \tag{15}$$

Finally, For the photodetector device electronic conductivity gain A is given by A = N$_t$ /($\Delta n$ +n$_1$). Under low-injection State $\Delta n \ll$ n$_1$. Therefore, A $\approx$ Nt/n$_1$ and noted that gain has the peak value. As the intensity of incident light is increased, $\Delta n$ progressively increases and gain A decreases. When the intensity of incident light is increased beyond, the gain is almost dead. Therefore, it is concluded that traps are maintained to support considerably high gain.

From the above derived mathematical model, it was clear that the traps which are created upon varying the density of surface states have considerably increasing the photogenerated current resulting in increased responsivity for nitrogen doped MoS$_2$/n-Si device compared to undoped MoS$_2$/n-Si device.

**Table 1. Comparison of responsivities values with existing literature.**

| Device | Detection Range | Responsivity (A/W) | Reference |
|---|---|---|---|
| MoS$_2$-BN-G | 400–885 nm | 180 | [31] |
| WSe$_2$-SnS$_2$ | 400–900 nm | 244 | [32] |
| ReSe$_2$ | 633nm | 95 | [29] |
| MoS$_2$ | 532nm | 59 | [33] |
| In$_2$Se$_3$ (PVD) | 532nm | 340 | [34] |
| GaTe | 254–710nm | 274.3 | [35] |
| G/WSe$_2$ | 532nm | 350 | [36] |
| MoS$_2$/SnS$_2$ QDs | UV, Visible, NIR | 435 | [37] |
| MoS$_2$ local Vbg | 532–638nm | 342.6 | [38] |
| MoS$_2$-ReS$_2$ | 555nm | 400 | [39] |
| MoS$_2$/TiO$_2$ nanoflowers | UV | 35.9 | [40] |
| Te NWs/ReS$_2$ | Visible light (632nm) | 180 | [41] |
| BaTiO$_3$nanoparticles/ MoS$_2$ | UV (365nm) | 120 | [42] |
| CuO NWs/MoS$_2$ | Visible light | 157.6 | [43] |
| **Nitrogen doped MoS$_2$** | **Visible range** | **606** | **This work** |

Table 1 shows the comparison of responsivities values with existing *literature. Quoc et al.* have fabricated vdWHs photodetector using MoS$_2$-BN-G and calculated responsivity as 180A/ W. *Xing Zhou et al.* have fabricated highly sensitive WSe$_2$/SnS$_2$ photodiode and examined responsivity to be 244 A/W. *S. Yang et al.* [33] have fabricated a single-layer ReSe$_2$ photo-transistor and examined responsivity to be 95 A/W. *Weiwei Tang et al. have* fabricated MoS$_2$-based FETs and noted responsivity as 59 A/W. *Jiadong Zhou et al.* have synthesised of high-quality monolayered In$_2$Se$_3$ FET's using physical vapor deposition and noted responsivity as 340 A/W. *Pingan Hu et al.* have fabricated GaTe nanosheet phototransistor and noted responsivity as 274.3 A/W. *Anyuan Gao et al.* have fabricated atomically thin graphene/WSe$_2$ heterojunctions and noted responsivity as 350A/W. *Chandra Sekhar Reddy Kolli et al.* have fabricated Monolayer MoS$_2$/SnS$_2$ Quantum Dot-Based Photodetector and observed responsivity to be 435A/W. *Junyeon Kwon et al.* have fabricated Multilayer MoS 2 Phototransistors and observed responsivity of 342.6A/W. *polumati et al.* have fabricated, MoS$_2$-ReS$_2$ heterostructure based photodetector and noted responsivity as 400 A/W. *Paul et al.* have fabricated photodetector using MoS$_2$/TiO$_2$ nanoflowers and measured responsivity as 36.9A/W. *Lee et al.* have fabricated a photodetctor using Te NWs/ReS$_2$ and noted responsivity as 180 A/W. *Ying et al.* have fabricated photodetector using BaTiO$_3$nanoparticles/ MoS$_2$ and measured responsivity as 120A/W. *Um et al.* fabricated photodetector using CuO NWs/MoS$_2$ and measured responsivity as 157.6 A/W.

This work explains the successful fabrication of nitrogen doped MoS$_2$ as a photodetector. For conventional photodetectors the responsivity calculated will be due internal electric field developed at the device interface. But, in this work the responsivity was further enhanced by improving conductivity of the device with nitrogen doping. By doping MoS$_2$ with nitrogen, density of surface states was considerably increased which contributes to additional conduction apart from that was obtained by internal electric field at device interface. As a result, the MoS$_2$ device conductivity was improved by varying surface states with nitrogen doping and noticeably enhanced responsivity. One of the potential advancements is this work is the observation that the surface states increase the responsivity of the photodetector, and that the responsivity of the photodetector can be tuned by surface states. The practical applications that could be achieved would be visible light communication, healthcare (SpO$_2$ monitoring), and

security applications (human motion monitoring). Further the challenges to achieve the practical applications are:

1. Controlled doping for controlled responsivity

2. As responsivity increases the response time tend to decrease. Hence to have a perfect balance of responsivity and response time is essential for practical applications

3. The growth of monolayer MoS$_2$ needs to be optimized for having controlled n-type behaviour (due to sulfur vacancies). The sulfur vacancies will decide the density of the surface states.

4. Finally the passivation of the device to have least interference with the environmental conditions which might affect the responsivity.

## Conclusion

In conclusion, this paper is all about enhancing the Responsivity of nitrogen doped CVD grown monolayer MoS$_2$ photodetector. The as fabricated device is experimentally studied and concluded that doping will tremendously vary density of surface states resulting in creating trap traps states well below the conduction band and these trap states are in turn responsible in capturing excess carriers. Also, for nitrogen doped MoS$_2$/n-Si device it was observed that density of states Nss increases proportionally to 16.3 X 10$^{12}$ states/m$^2$/eV for nitrogen doped MoS$_2$/n-Si device compared to undoped MoS$_2$/n-Si device of 4.2 x 10$^{12}$ states/m$^2$/eV. The variation in density of surface states creates traps and carriers that are being captured in these trap energy level for doped MoS$_2$/n-Si device contribute to additional photogenerated current upon light incidence which eventually a major responsible in increasing responsivity for about 10 times from undoped MoS$_2$/n-Si device to nitrogen doped MoS$_2$/n-Si device. It was noticed that due to presence of trap states for nitrogen doped MoS$_2$/n-Si device, the responsivity was noted to be 606 A/W when compared to undoped MoS$_2$/n-Si device having responsivity of only 65.12 A/W. Moreover, the devices having higher responsivity shows lower rise time of 0.35 sec and the device having low responsivity shows higher rise time of 0.85 sec. In addition to calculating density of states for both undoped MoS$_2$/n-Si and nitrogen doped MoS$_2$/n-Si devices at ambient temperature, experiment was also conducted to calculate density of states for nitrogen doped MoS$_2$/n-Si device with gradual increase in device temperatures.it was observed that with proportionate increase in device temperature the barrier height of the device is decreased and the reverse saturation current is increased and importantly the density of states and responsivity plots shows decrease in trend. This is due to fact that upon increase in device temperature, carriers that are captured in trap states are being excited to conduction band. As a result, these trap states are depleted of excess carriers resulting in decrease in density of states Nss and responsivity of the device.

## Supporting information

**S1 Fig.** a) FESEM images of nitrogen doped MoS$_2$. b) HRTEM image of monolayer MoS$_2$ c) XPS survey spectra of N-doped MoS$_2$, d), e) & f) Individual high-resolution XPS spectra of Mo 3d, S 2p & N 1s of nitrogen doped MoS$_2$.
(TIF)

**S2 Fig.** a) The Raman and b) PL spectra of monolayer MoS$_2$ that has been nitrogen-doped and undoped on a SiO2/Si substrate.
(TIF)

**S3 Fig.** a-b-c) UPS spectra (measured by He I source, hv = 21.22 eV) of n-type silicon substrate, undoped MoS$_2$ and nitrogen doped MoS$_2$ grown on n-type substrate. c)Schematic representation of energy-band diagram n-type of silicon substrate and undoped MoS$_2$ in equilibrium condition (when isolated) d) energy-band diagram n-type of silicon substrate and nitrogen doped MoS$_2$ in equilibrium condition (when isolated).
(TIF)

## Author Contributions

**Data curation:** Chandra Sekhar Reddy Kolli.

**Investigation:** Chandra Sekhar Reddy Kolli.

**Methodology:** Gowtham Polumati, Chandra Sekhar Reddy Kolli.

**Resources:** Parikshit Sahatiya.

**Supervision:** Andres de Luna Bugallo, Parikshit Sahatiya.

**Validation:** Andres de Luna Bugallo.

**Writing – original draft:** Gowtham Polumati.

**Writing – review & editing:** Andres de Luna Bugallo, Parikshit Sahatiya.

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
