## [Decision Letter · Decision Letter 0]

31 Oct 2023

PONE-D-23-33109Engineering Surface State Density of Monolayer CVD grown 2D MoS2 for Enhanced Photodetector PerformancePLOS ONE

Dear Dr. de Luna Bugallo,

Thank you for submitting your manuscript to PLOS ONE. After careful consideration, we feel that it has merit but does not fully meet PLOS ONE’s publication criteria as it currently stands. Therefore, we invite you to submit a revised version of the manuscript that addresses the points raised during the review process.

We look forward to receiving your revised manuscript.

Kind regards,

Niravkumar Joshi

Academic Editor

PLOS ONE

Journal Requirements:

   "P.S. acknowledges the funding from SERB (SRG/2020/ 000098). The authors thank Central Analytical Laboratory, BITS Pilani Hyderabad Campus, for the aid in material characterization.A part of the reported work (fabrication/characterization) was carried out at the lITBNF, IITB under INUP which is sponsored by DeitY, MCIT, Government of India."

3. We note you have included a table to which you do not refer in the text of your manuscript. Please ensure that you refer to Table 1 in your text; if accepted, production will need this reference to link the reader to the Table.

Reviewers' comments:

Reviewer's Responses to Questions

**Comments to the Author**

1. Is the manuscript technically sound, and do the data support the conclusions?

Reviewer #1: Yes

Reviewer #2: Yes

Reviewer #3: Yes

Reviewer #4: Yes

2. Has the statistical analysis been performed appropriately and rigorously? 

Reviewer #1: Yes

Reviewer #2: I Don't Know

Reviewer #3: Yes

Reviewer #4: Yes

3. Have the authors made all data underlying the findings in their manuscript fully available?

Reviewer #1: Yes

Reviewer #2: Yes

Reviewer #3: Yes

Reviewer #4: Yes

4. Is the manuscript presented in an intelligible fashion and written in standard English?

Reviewer #1: Yes

Reviewer #2: Yes

Reviewer #3: Yes

Reviewer #4: Yes

5. Review Comments to the Author

Reviewer #1: Bugallo et. al. reported the Engineering Surface State Density of Monolayer CVD grown 2D MoS2 for Enhanced Photodetector Performance. In order to verify dependence of surface states on the responsivity, author observed the surface states were varied by varying temperature and it was observed that upon increment in temperature, the surface states decreases which causes the responsivity values also to decrease. However, a few more clarifications are needed before the manuscript can be considered suitable for publication. A list of other comments that need to be addressed is as follows:

1. In abstract, Author should provide a more concise summary of the key findings and their significance. Avoid introducing new terms or concepts that are not explained in the abstract.

2. Expand the literature review to include more recent research in the field. Consider discussing the latest advancements in photodetector technology and 2D materials, which can highlight the relevance of your study.

3. The manuscript lacks graphical representations of data.

4. The discussion section is relatively brief. Consider expanding on the implications of your findings and how they contribute to the broader understanding of photodetectors.

5. Author should discuss the practical applications and potential advancements enabled by your research.

Reviewer #2: The author showed the effect of N doping in MoS2. The mathematical models seem adequate but paper lacks explanation of the measurement techniques in detail. It would have been good to compare experiment result with mathematical model.

Reviewer #3: 1. What is the role of surface states in enhancing the responsivity of photodetector devices, especially when doped with nitrogen?

2. How does varying the density of surface states through nitrogen doping affect the performance of a monolayer MoS2 photodetector?

3. What are the practical applications for photodetectors with significantly enhanced responsivity, such as the nitrogen-doped MoS2 device described in the research?

4. What are the implications of the temperature-dependent nature of surface states on the responsivity of photodetector devices, and how can this be optimized for specific applications?

5. How does the presence of traps, created by variations in surface states, impact the recombination of carriers in the photodetector and its overall performance?

6. Can the findings and techniques described in this research be extended to improve the responsivity of photodetector devices using other materials or in different wavelength ranges?

7. What challenges and limitations might researchers encounter when attempting to implement the methods described in this study for practical device applications?

Reviewer #4: The manuscript is well written and supported by good experimental and modeling data. Following comments need to be addressed – Please see attached file. Thank you..............................................................................................................

6. PLOS authors have the option to publish the peer review history of their article (what does this mean?). If published, this will include your full peer review and any attached files.

Reviewer #1: No

Reviewer #2: **Yes: **Nayan Chakravarty

Reviewer #3: No

Reviewer #4: No

---

## [Author Response · Author response to Decision Letter 0]

24 Nov 2023

November 21st, 2023

RE: Revision Requested for Manuscript reference: PONE-D-23-33109.

Dear Editor,

Thank you for your email dated 31-October-2023 communicating the review on our manuscript reference: PONE-D-23-33109 titled. " Engineering Surface State Density of Monolayer CVD grown 2D MoS2 for Enhanced Photodetector Performance". We are glad that reviewers found the manuscript publishable after a revision. We want to thank you and the reviewers for giving our manuscript an opportunity to be considered by PLOS ONE. We also appreciate the editors and reviewer’s time and insight comments that have helped us in improving the manuscript. Below please find a detailed delineation of our responses to the comments.

The document attached at the submersion page. Thank You!

---

## [Decision Letter · Decision Letter 1]

15 Jan 2024

Engineering Surface State Density of Monolayer CVD grown 2D MoS2 for Enhanced Photodetector Performance

PONE-D-23-33109R1

Dear Dr. Andres,

We’re pleased to inform you that your manuscript has been judged scientifically suitable for publication and will be formally accepted for publication once it meets all outstanding technical requirements.

Kind regards,

Niravkumar Joshi

Academic Editor

PLOS ONE

Additional Editor Comments (optional):

Reviewers' comments:

Reviewer's Responses to Questions

**Comments to the Author**

1. If the authors have adequately addressed your comments raised in a previous round of review and you feel that this manuscript is now acceptable for publication, you may indicate that here to bypass the “Comments to the Author” section, enter your conflict of interest statement in the “Confidential to Editor” section, and submit your "Accept" recommendation.

Reviewer #1: All comments have been addressed

Reviewer #2: All comments have been addressed

2. Is the manuscript technically sound, and do the data support the conclusions?

Reviewer #1: Yes

Reviewer #2: Yes

3. Has the statistical analysis been performed appropriately and rigorously? 

Reviewer #1: Yes

Reviewer #2: Yes

4. Have the authors made all data underlying the findings in their manuscript fully available?

Reviewer #1: Yes

Reviewer #2: Yes

5. Is the manuscript presented in an intelligible fashion and written in standard English?

Reviewer #1: Yes

Reviewer #2: Yes

6. Review Comments to the Author

Reviewer #1: Conclusion on research article:

Bugallo et. al. reported the Engineering Surface State Density of Monolayer CVD grown 2D MoS2 for Enhanced Photodetector Performance. All the comments are addressed properly, that manuscript is ready for publication.

Reviewer #2: The Author addressed all the questions. No further questions from my side. Thanks ....................

7. PLOS authors have the option to publish the peer review history of their article (what does this mean?). If published, this will include your full peer review and any attached files.

Reviewer #1: No

Reviewer #2: No

---

## [Editor Report · Acceptance letter]

1 Feb 2024

PONE-D-23-33109R1 

PLOS ONE

Dear Dr. de Luna Bugallo, 

I'm pleased to inform you that your manuscript has been deemed suitable for publication in PLOS ONE. Congratulations! Your manuscript is now being handed over to our production team.

Kind regards, 

on behalf of

Dr. Niravkumar Joshi 

Academic Editor

PLOS ONE